# Thermo-responsive chiral micelles as recyclable organocatalyst for asymmetric Rauhut-Currier reaction in water

Lei Xu[1,2,5], Li Zhou[3,5], Yan-Xiang Li[3,5], Run-Tan Gao[1], Zheng Chen [1], Na Liu[4] & Zong-Quan Wu [1]✉

Developing eco-friendly chiral organocatalysts with the combined advantages of homogeneous catalysis and heterogeneous processes is greatly desired. In this work, a family of amphiphilic one-handed helical polyisocyanides bearing phosphine pendants is prepared, which self-assembles into well-defined chiral micelles in water and showed thermo-responsiveness with a cloud point of approximately 38.4 °C. The micelles with abundant phosphine moieties at the interior efficiently catalyze asymmetric cross Rauhut-Currier reaction in water. Various water-insoluble substrates are transferred to target products in high yield with excellent enantioselectivity. The yield and enantiomeric excess (ee) of the product generated in water are up to 90% and 96%, respectively. Meanwhile, the yields of the same R-C reaction catalyzed by the polymer itself in organic solvents is <16%, with an ee < 72%. The homogeneous reaction of the chiral micelles in water turns to heterogeneous at temperatures higher than the cloud point, and the catalyst precipitation facilitates product isolation and catalyst recovery. The polymer catalyst is recycled 10 times while maintaining activity and enantioselectivity.

One of the important goals in catalysis is the development of eco-friendly catalysts with the combined advantages of homogeneous catalysis and heterogeneous processes, which not only maintain or even improve the catalytic activity and selectivity of homogeneous catalysts but also facilitate product isolation and catalyst recycling[1–4]. Homogeneous catalysts are widely used in fine-chemical synthesis because typical solid-supported heterogeneous catalysts do not provide the nonpolar environments often required for organic reactions. Soluble polymers are less routinely used catalyst supports that could provide a solvent-like environment for organic reactions[5–9]. Therefore, polymer skeletons to support catalysts that can increase both catalytic activity and selectivity are greatly desired, especially for chiral catalysts utilized in asymmetric reactions[10,11].

Homochirality is one of the most remarkable features of biological molecules[12]. Biopolymers can express their homochirality by twisting into one-handed helices (e.g., the α-helix of proteins and the double helix of DNA)[13,14]. Enzyme-catalyzed stereospecific reactions are believed to arise from the homochirality of macromolecular helix[15,16]. Inspired by such helices of biomacromolecules, artificial helical polymers have attracted great research attention because of not only their unique structures but also their broad applications, such as chiral recognition and resolution, circularly polarized luminescence, and so forth[17–30]. Helical polymers are good skeletons to support chiral organocatalysts because helical backbones can provide additional chiral microenvironments, and improve the stereoselectivity of an asymmetric reaction[31,32]. Helicity itself could

[1]State Key Laboratory of Supramolecular Structure and Materials, College of Chemistry, Jilin University, 130012 Changchun, China. [2]Key Laboratory of Green and Precise Synthetic Chemistry and Applications, Ministry of Education, Huaibei Normal University, 235000 Huaibei, Anhui, China. [3]Department of Polymer Science and Engineering, Hefei University of Technology, 230009 Hefei, China. [4]The School of Pharmaceutical Sciences, Jilin University, 1266 Fujin Road, 130021 Changchun, Jilin, China. [5]These authors contributed equally: Lei Xu, Li Zhou, Yan-Xiang Li. ✉e-mail: zqwu@jlu.edu.cn

induce the high enantioselectivity of some asymmetric reactions[33,34]. Reversing helicity can switch enantioselectivity, thus allowing the obtaining of enantiomeric products[35,36]. Moreover, the high molecular weight of helical polymers can simplify product isolation and facilitate catalyst recycling, which are particularly desirable for expensive and hardly available chiral catalysts[10,37–39]. In this respect, polyisocyanide is one of the most attractive helical polymers because of its unique rigid rod-like backbone, high stability, and good self-assembly tendency[17–21,25]. Therefore, it is a good skeleton for fabricating chiral catalysts for asymmetric reactions.

Water is the cheapest and the most environmentally friendly solvent. As organic compounds are generally nonpolar and water-insoluble, organic reactions in water are commonly restricted[40]. However, enzymes perform catalytic reactions in aqueous systems with high efficiency and excellent selectivity[41,42]. On the basis of the understanding of enzyme catalysis, polymer-based chiral catalysts have been explored[43–46]. In contrast, the knowledge about organocatalytic chiral micelles for asymmetric reactions in water with high enantioselectivity and efficiency is still in its infancy. During the past decades, asymmetric organocatalysis has gained great attention because of its advantages, including inexpensive and easily available catalysts, no metal residues, and mild reaction conditions[47–49]. The Rauhut–Currier (R–C) reaction of two active olefins is a unique and efficient approach for constructing carbon-carbon bonds and densely functionalized organic building blocks[50–54]. Phosphine-catalyzed intermolecular cross R–C reaction is particularly intriguing among various organocatalyzed reactions[52]. Moreover, because of the poor solubility of reactants and limited catalysts, efficient cross R–C reaction in water with high enantioselectivity has not been realized to date.

We herein describe the construction of chiral organocatalytic micelles using amphiphilic helical polyisocyanide copolymers, composed of hydrophobic helical polyisocyanide bearing phosphine pendants and hydrophilic polyisocyanide carrying methyl triglycol chains. In water, the polymers self-assembled into well-defined chiral micelles with the hydrophobic phosphine pendants at the interior. The micelles catalyzed the asymmetric cross R–C reaction of various water-insoluble substrates in water and yielded the desired products in high yields with excellent enantioselectivity. The enantiomeric excess (ee) and yield of the product were up to 96% and 90%, respectively. Moreover, the block copolymers had excellent thermo-responsiveness in water with a cloud point of 38.4 °C. The precipitation of polymers at temperatures higher than the cloud point facilitated product isolation and catalyst recycling. The polymer catalyst was recycled 10 times with maintained activity and enantioselectivity.

## Results

### Polymer synthesis and characterization

The block copolymers were prepared following Fig. 1a. Chiral isocyanides (**1r** and **1s**) bearing boron hydride-protected phosphine were polymerized by an alkyne-Pd(II) catalyst and gave the desired polymers in high yield with predicted molar mass ($M_n$) and low dispersity ($M_w/M_n$)[17]. For example, the $M_n$ and $M_w/M_n$ of poly-**1s**$_{50}$ (the footnote indicates the initial monomer-to-catalyst feed ratio, as below) were 23.1 kDa and 1.22, respectively, as determined by size exclusion chromatography (SEC) (Fig. 1b). Because polymerization follows a living polymerization mechanism, poly-**1s**$_{50}$ bearing an active Pd(II)-complex on the chain end was chain extended with achiral isocyanide (**2**) bearing methyl triglycol chains[17]. The $M_n$ of the resulting poly(**1s**$_{50}$-b-**2**$_{100}$) copolymer was

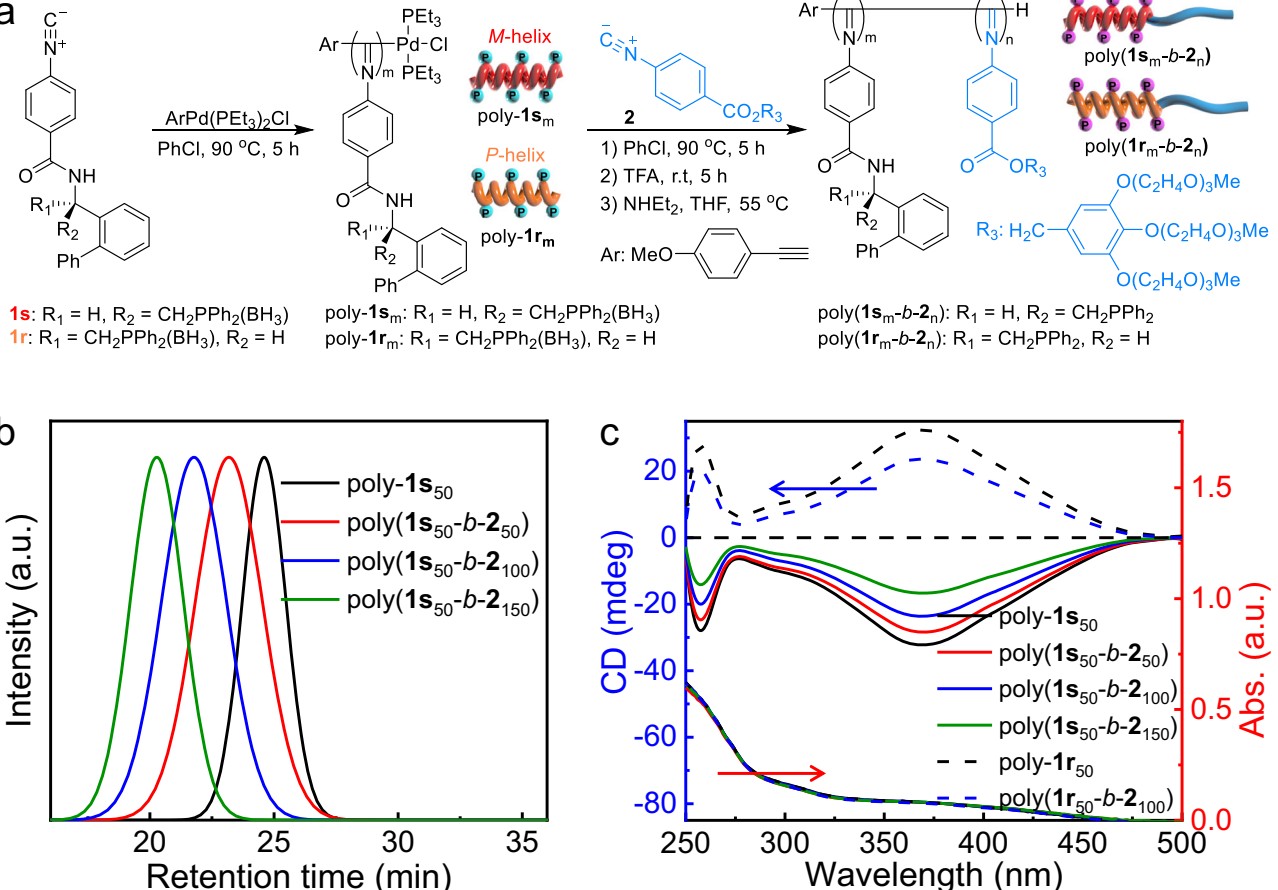

**Fig. 1 | Synthesis of helical polymers. a** Synthetic route for polyisocyanide block copolymers. **b** Size-exclusion chromatograms (eluent: THF, the a.u. is the abbreviation of arbitrary units), and **c** CD and UV–vis spectra of the synthetic polymers (0.2 mg/mL, THF, 25 °C).

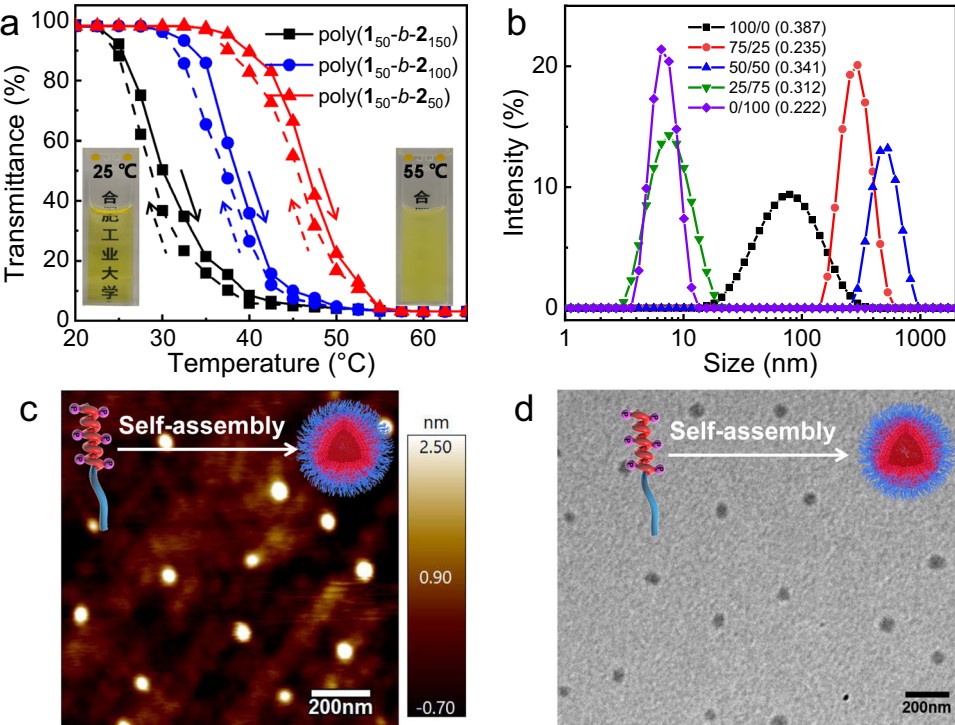

**Fig. 2 | Thermo-responsive and self-assembly properties. a** Plots of the transmittance of poly($1s_{50}$-$b$-$2_{100}$) in water versus temperature (the solid and dashed lines represent the heating and cooling process, respectively). Insets: photographs of poly($1s_{50}$-$b$-$2_{100}$) in $H_2O$ at 25 and 55 °C, 1.0 mg/mL). **b** DLS traces for poly($1s_{50}$-$b$-$2_{100}$) in the mixture of $H_2O$ and THF with different volume ratios (0.2 mg/mL). The polydispersities for the DLS analyses are 0.387 ($H_2O$), 0.235 ($H_2O$/THF = 75/25), 0.341 ($H_2O$/THF = 50/50), 0.312 ($H_2O$/THF = 25/75), and 0.222 (THF), respectively. **c** AFM and **d** TEM images of poly($1s_{50}$-$b$-$2_{100}$) casted from the aqueous solution at room temperature.

84.2 kDa, and it maintained a low dispersity with $M_w/M_n = 1.25$ (Fig. 1b). The afforded block copolymer was successively treated with triethylphosphine and trifluoroacetic acid to remove the boron hydride that protected the phosphine pendants and the Pd(II)-terminal. Then, poly($1s_m$-$b$-$2_n$) copolymers with different compositions and various block ratios were prepared using the living nature of polymerization (Fig. 1b and Supplementary Table 1). Similarly, poly($1r_m$-$b$-$2_n$) copolymers using enantiomeric $1r$ instead of $1s$ were also prepared (Supplementary Table 1). Apart from SEC, these polymers were characterized by $^1H$ and $^{31}P$ NMR and FT-IR (Supplementary Figs. 1–9). Because the polymerizations proceeded in a well-controlled living polymerization mechanism, the degree of polymerization was consistent with the initial monomer-to-catalyst feed ratio, according to the detailed studies we reported previously[11,17,55,56].

The helicity of the prepared polymers was studied using circular dichroism (CD) spectroscopy in tetrahydrofuran (THF) at 25 °C. Because of the asymmetric induction of the chiral monomer, poly-$1s_{50}$ showed an intense negative CD in the absorption region of the polyisocyanide backbone, suggesting that the backbone was twisted into a left-handed helix (Fig. 1c)[55,56]. After chain extension, the resulting poly($1s_{50}$-$b$-$2_n$) showed a negative CD similar to that of the poly-$1s_{50}$ macroinitiator, whereas the recorded molecular CD intensity was decreased (Fig. 1c). Detailed analyses revealed that the molecular CD intensity of the poly-$1s_m$ segment was maintained during block copolymerization (Supplementary Fig. 10). The CD decrease was ascribed to $2$ being achiral, and the resulting poly-$2_n$ segment could not maintain one-handed helicity; thus, the entire molecularly optical activity was decreased. This study confirmed that the poly-$1s_m$ segment bearing catalytic phosphine pendants of the block copolymers adopted left-handed helicity. The helicity was quite stable, and no obvious change could be detected in various solvents at temperatures from 5 to 60 °C (Supplementary Fig. 11). As anticipated, the poly-$1r_m$ segment of the

poly($1r_m$-$b$-$2_n$) copolymers adopted right-handed helicity, as revealed by CD and adsorption spectroscopy techniques (Fig. 1c and Supplementary Fig. 12)[55,56].

Because of their amphiphilic character, the block copolymers could be dissolved in various organic solvents and in water. Interestingly, the transparent aqueous solution of poly($1s_{50}$-$b$-$2_{100}$) turned turbid upon heating and became transparent again after cooling to room temperature, suggesting the turbidimetry-responsiveness with temperature (Fig. 2a). Detailed UV–vis absorption studies revealed that the cloud point was 38.4 °C for poly($1s_{50}$-$b$-$2_{100}$), determined from the temperature corresponding to 50% transmittance of the antisigmoidal transmittance–temperature curve during the heating process[57]. The cloud point decreased with the elongation of poly-$2_n$ block of the copolymers, it was 46.7, 38.4, and 30.0 °C for poly($1s_{50}$-$b$-$2_{50}$), poly($1s_{50}$-$b$-$2_{100}$), and poly($1s_{50}$-$b$-$2_{150}$), respectively (Fig. 2a).

The self-assembly property of the block copolymers was investigated by adding water to their THF solutions. Dynamic light scattering (DLS) analyses indicated that poly($1s_{50}$-$b$-$2_{100}$) had a hydrodynamic diameter of ca. 8 nm in THF, suggestive of molecular dissolution (Fig. 2b). After adding water, the diameters were 506 and 280 nm for the water contents of 50% and 75%, respectively, suggesting that the amphiphilic block copolymer was self-assembled into micelles with hydrophilic poly-$2_{100}$ at the exterior and hydrophobic poly-$1s_{50}$ at the interior. In pure water, the diameter further decreased to 85 nm, indicating that poly($1s_{50}$-$b$-$2_{100}$) was self-assembled into a more compact micelle (Fig. 2b). The critical aggregation concentration (CAC) in water was as low as 0.040 mg/mL, suggesting that this polymer had good self-assembly property (Supplementary Fig. 13). The morphology of the micelles was confirmed by atomic force microscopy (AFM) and transmission electron microscopy (TEM). As displayed in Fig. 2c, the AFM phase image of poly($1s_{50}$-$b$-$2_{100}$) cast from the aqueous solution showed spherical nanoparticles in good homogeneity with a diameter of 75 nm.

**Table 1 | Optimization of the R–C reaction condition[a]**

| Run | Catalyst | X[b] | Solution | Temp. (°C) | Yield (%)[c] | ee (%)[d] |
|---|---|---|---|---|---|---|
| 1 | poly($1s_{50}$-b-$2_{100}$) | 4 | CHCl$_3$ | 25 | 16 | 70 |
| 2 | poly($1s_{50}$-b-$2_{100}$) | 4 | Toluene | 25 | 14 | 55 |
| 3 | poly($1s_{50}$-b-$2_{100}$) | 4 | Acetone | 25 | 12 | 65 |
| 4 | poly($1s_{50}$-b-$2_{100}$) | 4 | EtOH | 25 | 15 | 50 |
| 5 | poly($1s_{50}$-b-$2_{100}$) | 4 | THF | 25 | 11 | 72 |
| 6 | poly($1s_{50}$-b-$2_{100}$) | 4 | THF/H$_2$O (75/25) | 25 | 20 | 75 |
| 7 | poly($1s_{50}$-b-$2_{100}$) | 4 | THF/H$_2$O (50/50) | 25 | 43 | 80 |
| 8 | poly($1s_{50}$-b-$2_{100}$) | 4 | THF/H$_2$O (25/75) | 25 | 61 | 84 |
| 9 | poly($1s_{50}$-b-$2_{100}$) | 4 | H$_2$O[e] | 25 | 84 | 90 |
| 10 | poly($1s_{50}$-b-$2_{50}$) | 4 | H$_2$O[e] | 25 | 83 | 79 |
| 11 | poly($1s_{50}$-b-$2_{150}$) | 4 | H$_2$O[e] | 25 | 85 | 87 |
| 12 | poly($1s_{50}$-b-$2_{100}$) | 2 | H$_2$O[e] | 25 | 56 | 84 |
| 13 | poly($1s_{50}$-b-$2_{100}$) | 8 | H$_2$O[e] | 25 | 79 | 81 |
| 14 | poly($1s_{50}$-b-$2_{100}$) | 10 | H$_2$O[e] | 25 | 84 | 82 |
| 15 | poly($1s_{50}$-b-$2_{100}$) | 4 | H$_2$O[e] | 15 | 83 | 92 |
| 16 | poly($1s_{50}$-b-$2_{100}$) | 4 | H$_2$O[e] | 5 | 82 | 94 |
| 17 | poly($1s_{50}$-b-$2_{100}$) | 4 | H$_2$O[e] | 0 | 81 | 96 |
| 18 | poly($1r_{50}$-b-$2_{100}$) | 4 | H$_2$O[e] | 0 | 82 | 95 (S) |
| 19 | poly-$1s_{50}$ | 4 | H$_2$O[e] | 0 | 51 | 69 |
| 20 | **1s** | 4 | H$_2$O[e] | 0 | 57 | 48 |
| 21 | **1r** | 4 | H$_2$O[e] | 0 | 55 | 49 (S) |
| 22 | poly-$1s_{50}$ | 4 | CHCl$_3$ | 0 | 82 | 76 |
| 23 | **1s** | 4 | CHCl$_3$ | 0 | 78 | 57 |

[a]Unless otherwise specified, all reactions were carried out with **3a** (0.1 mmol) and **4a** (0.3 mmol) in given solvent (5 mL).
[b]The loading of the catalyst was determined by elemental analysis.
[c]Yield of isolated products.
[d]Determined by HPLC analysis using a chiral column.
[e]<5 volume % of THF were used in case the substrates could not be dissolved.

Meanwhile, TEM images further supported that the block copolymer was self-assembled into core-shell-like micelles with a diameter of 72 nm (Fig. 2d). The relatively large size of the micelles was ascribed to the formation of hollowed spherical micelles because of the distinct rigid and rod-like backbone of polyisocyanides[58,59]. The hollowed micelles might facilitate substrate exchange during the following asymmetric R–C reaction. The cryo-TEM image of poly($1s_{50}$-b-$2_{100}$) in water also supported the formation of spherical micelles with a diameter of ca. 90 nm (Supplementary Fig. 14). Other block copolymers showed similar self-assembly properties in water, as revealed by the DLS analyses (Supplementary Fig. 15). Accordingly, poly($1r_{50}$-b-$2_{100}$), possessing opposite handed helicity, showed a similar self-assembly behavior (Supplementary Fig. 16).

**Asymmetric cross R–C reactions**
The catalytic activity of poly($1s_m$-b-$2_n$) micelles for intermolecular cross R–C reactions was explored using ethyl (E)-4-oxo-4-phenylbut-2-enoate (**3a**) and but-3-en-2-one (**4a**) as model substrates. Initially, the reaction was conducted at room temperature with 10 mol% catalyst loading of the phosphine pendants of poly($1s_{50}$-b-$2_{100}$) in various organic solvents for 48 h. Then, homogeneous reactions occurred and gave the target product R-**5aa**. However, the reaction efficiency was

quite low, and the yields of R-**5aa** were only approximately 16% (runs 1–5, Table 1). The homogeneous cross-R–C reaction in THF and CHCl$_3$ could give the target product R-**5aa**, but the yield and ee values were not satisfactory (runs 1 and 5, Table 1). The ee of R-**5aa** was generally <72%, as determined by high-performance liquid chromatography (HPLC) using a chiral column (see the Supplementary Information for more details). Then, we conducted the reaction in a mixture of THF and water with different volume ratios. We found that both the reaction rate and enantioselectivity were improved with the addition of water to THF (Fig. 3a and Supplementary Fig. 17). For example, the yields of R-**5aa** were 20%, 43%, and 61% with the addition of 25%, 50%, and 75% water to the THF solution, respectively (runs 6–8, Table 1). In pure water, the isolated yield of R-**5aa** was higher than 84%. The enantioselectivity of the reactions showed the same tendency. As plotted in Fig. 3b, the ee values of the generated R-**5aa** were 75%, 80%, and 84% for the reactions conducted in mixtures of THF and water with 25%, 50%, and 75% water content at room temperature, respectively. As expected, the reaction in water catalyzed by poly($1s_{50}$-b-$2_{100}$) showed the best enantioselectivity, the ee of the target was as high as 90%. The polymer catalysts carried chiral carbon centers on the pendants and possessed a chiral helical backbone. Thus, to obtain details on enantioselectivity, asymmetric R–C reaction of **3a** with **4a**

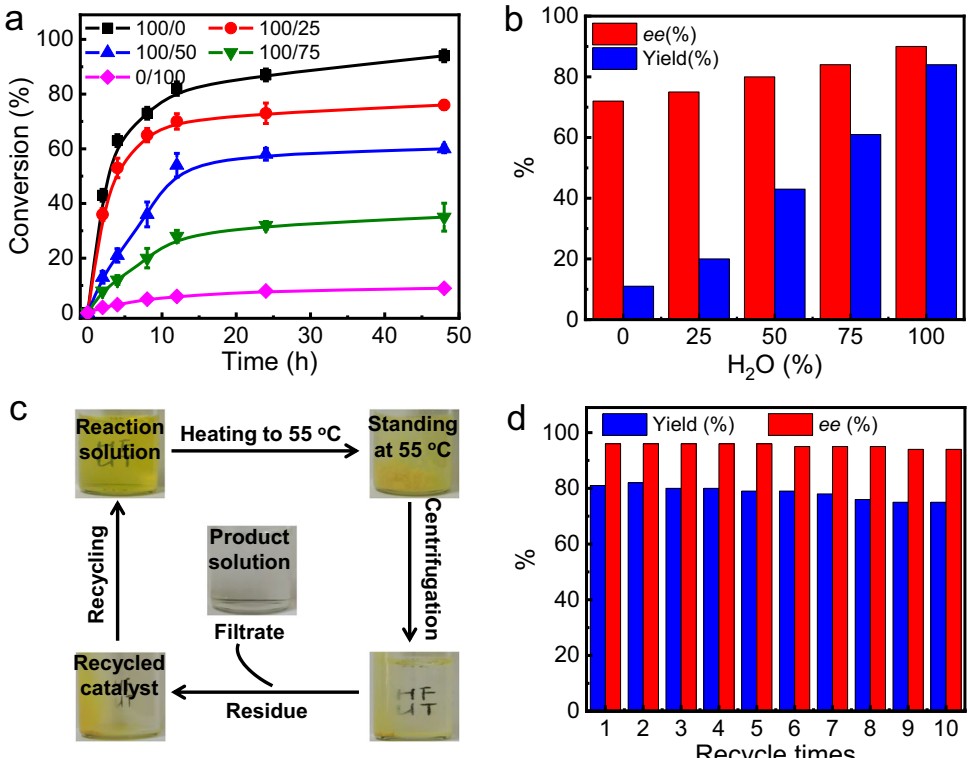

**Fig. 3 | Results for the polymer micelle catalyzed cross R−C reaction. a** Plot of conversion of **3a** versus reaction time catalyzed by poly(**1s**$_{50}$-*b*-**2**$_{100}$) in different H$_2$O/THF ratios. Error bars of measured conversion versus reaction time. **b** The yield and *ee* of ***R*-5aa** generated in different ratios of H$_2$O/THF using poly(**1s**$_{50}$-*b*-**2**$_{100}$) as catalyst. **c** Photographs of the poly(**1s**$_{50}$-*b*-**2**$_{100}$) catalyst recycling. **d** The results for recycling poly(**1s**$_{50}$-*b*-**2**$_{100}$) in the reaction of **3a** and **4a**.

catalyzed by the poly-**1s**$_{50}$ homopolymer, **1r** and **1s** monomers were conducted under identical conditions (runs 19–23, Table 1). The reactions in water gave **5aa** in 51% yield with 69% ee using poly-**1s**$_{50}$ as the catalyst, whereas it gave **5aa** in 57% yield with 48% ee using **1s** as the catalyst. The homogeneous reactions in CHCl$_3$ gave the desired **5aa** in 82% yield and 76% ee using poly-**1s**$_{50}$; and 78% yield and 57% ee using **1s**. The **1r** showed behaviors similar to those of **1s** but with opposite enantioselectivity (run 21, Table 1). The relatively higher yield and ee values obtained by poly-**1s**$_{50}$ than those of **1s** confirmed the synergistic effect of the helical backbone and the chiral pendants. In addition, an increase in $M_n$ of poly-**1s**$_n$ could further improve the enantioselectivity of **5aa** until the degree of the polymerization of poly-**1s**$_n$ reached 50 (Supplementary Fig. 18a). These results indicated the $M_n$-dependent helicity of the poly-**1s**$_n$ backbone and further supported that the enhanced enantioselectivity came from the helical chirality of the polymer catalyst[56]. Considering the local concentration of substrates within the micelle might influence the catalysis, experiments at different substrate concentrations were performed. The best results were obtained using 0.1 mmol of **3a** with 0.3 mmol of **4a** in water (5 mL) (Supplementary Fig. 18b). A further increase in concentration caused precipitation, whereas the dilution of the substrates gave the product in lower yield. Note that <5% volume of THF was used in case the substrates could not be dissolved in water.

The results inspired us to investigate the effect of catalyst composition on reactions. The cross R−C reaction catalyzed by poly(**1s**$_{50}$-*b*-**2**$_{50}$), poly(**1s**$_{50}$-*b*-**2**$_{100}$), and poly(**1s**$_{50}$-*b*-**2**$_{150}$) was performed in water under the same conditions. Therein, poly(**1s**$_{50}$-*b*-**2**$_{100}$) showed the best results in terms of the yield and ee of the target product (runs 9–11, Table 1). Thus, this polymer was applied in the following studies. Because poly(**1s**$_{50}$-*b*-**2**$_{100}$) had good solubility in water, the reaction of **3a** and **4a** was performed in water at 0 °C. As summarized in Table 1, the reaction gave the expected ***R*-5aa** in 81%

yield and 96% ee. Because the CAC of the block copolymer was quite low, the loading of the block polymer catalyst was further decreased to 2 mol% of phosphine pendants; however, both the reaction rate and enantioselectivity decreased considerably (run 12, Table 1). According to these studies, the optimized conditions were carrying the R−C reaction in water at 0 °C with 4 mol% loading of the catalyst (based on the phosphine). Given these results, the intermolecular cross R−C reaction was conducted using the poly(**1r**$_{50}$-*b*-**2**$_{100}$) catalyst possessing the opposite, right-handed helical backbone under the same conditions described above. Gratifyingly, the reaction of **3a** with **4a** delivered the desired enantiomeric product ***S*-5aa** in 82% yield and 95% ee. These results suggest that the enantioselectivity of the R−C reaction could be reversed by tuning the helicity of the polymer backbone.

The aforementioned results encouraged us to explore the substrate scope of the R−C reaction. Thus, 2-ene-1,4-diones and vinyl ketones with different substituents were prepared and applied in an asymmetric cross-R−C reaction using the poly(**1s**$_{50}$-*b*-**2**$_{100}$) catalyst. As shown in Table 2, the catalyst was applicable to a wide range of 3-aroyl acrylates (**3b**–**3h**) with different aryl substituents. The targeted products with high yields and excellent enantioselectivities were attained regardless of the electron-donating or electron-withdrawing substituents (runs 1–7, Table 2). Notably, when the ethyl ester group of **3a** was replaced by the less hindered methyl ester (**3i**) or the more hindered isopropyl ester (**3j**) and benzyl ester (**3k**), the desired products (**5ia**–**5ka**) were also obtained with good yield (82–85%) and excellent enantioselectivity (95–96% ee) (8–10, Table 2). Encouraged by these results, the helical polymer-based catalyst was applied to the cross-R−C reaction using aryl-substituted vinyl ketones (**4b**–**4d**). Gratifyingly, these ketones could also react with various 3-aroyl acrylates (**3g** and **3h**) catalyzed by poly(**1s**$_{50}$-*b*-**2**$_{100}$) in water and gave the expected products in good yields (69–72%) with high ee (90–93%)

**Table 2 | Scope of the enantioselective R–C reaction catalyzed by poly(1s$_{50}$-$b$-2$_{100}$)$^a$**

| Run | R$_1$/R$_2$ (3) | R$_3$ (4) | 5 | Yield (%)$^b$ | ee (%)$^c$ |
|---|---|---|---|---|---|
| 1 | 4-F-C$_6$H$_4$/Et (**3b**) | CH$_3$ (**4a**) | **5ba** | 87 | 93 |
| 2 | 4-Cl-C$_6$H$_4$/Et (**3c**) | CH$_3$ (**4a**) | **5ca** | 90 | 94 |
| 3 | 4-Br-C$_6$H$_4$/Et (**3d**) | CH$_3$ (**4a**) | **5da** | 86 | 95 |
| 4 | 4-Me-C$_6$H$_4$/Et (**3e**) | CH$_3$ (**4a**) | **5ea** | 80 | 96 |
| 5 | 4-MeO-C$_6$H$_4$/Et (**3f**) | CH$_3$ (**4a**) | **5fa** | 84 | 95 |
| 6 | 4-Ph-C$_6$H$_4$/Et (**3g**) | CH$_3$ (**4a**) | **5ga** | 81 | 96 |
| 7 | 2-naphthyl/Et (**3h**) | CH$_3$ (**4a**) | **5ha** | 83 | 96 |
| 8 | C$_6$H$_5$/Me (**3i**) | CH$_3$ (**4a**) | **5ia** | 85 | 95 |
| 9 | C$_6$H$_5$/iPr (**3j**) | CH$_3$ (**4a**) | **5ja** | 83 | 95 |
| 10 | C$_6$H$_5$/Bn (**3k**) | CH$_3$ (**4a**) | **5ka** | 82 | 96 |
| 11 | 4-Ph-C$_6$H$_4$/Et (**3g**) | 4-Me-C$_6$H$_4$ (**4b**) | **5gb** | 70 | 92 |
| 12 | 2-naphthyl/Et (**3h**) | 4-Me-C$_6$H$_4$ (**4b**) | **5hb** | 72 | 90 |
| 13 | 4-Ph-C$_6$H$_4$/Et (**3g**) | 4-F-C$_6$H$_4$ (**4c**) | **5gc** | 69 | 91 |
| 14 | 2-naphthyl/Et (**3h**) | 4-F-C$_6$H$_4$ (**4c**) | **5hc** | 70 | 90 |
| 15 | 2-naphthyl/Et (**3h**) | 4-Cl-C$_6$H$_4$ (**4d**) | **5hd** | 71 | 93 |
| 16 | 2-thienyl/Et (**3l**) | CH$_3$ (**4a**) | **5la** | 74 | 88 |
| 17 | 2-Furyl/Et (**3m**) | CH$_3$ (**4a**) | **5ma** | 76 | 84 |
| 18 | tBu/Et (**3n**) | CH$_3$ (**4a**) | **5na** | 68 | 89 |

$^a$Unless otherwise specified, all reactions were carried out with **3** (0.1 mmol) and **4** (0.3 mmol) in water (5 mL, <5 volume % of THF were used in case the substrates could not be dissolved in water), and the catalyst loading was determined by elemental analysis.
$^b$Yield of isolated products.
$^c$Determined by HPLC analysis using a chiral column.

(runs 11–15, Table 2). Furthermore, the substrate scope exploration suggested that poly(1s$_{50}$-$b$-2$_{100}$) was also an efficient catalyst for substrates containing heteroaryl groups and aliphatic chains. For example, the reaction of **3l**, **3m**, and **3n** with **4a** gave the expected products **5la**, **5ma**, and **5na** in good yield and high enantioselectivity (runs 16–18, Table 2). Collectively, these studies revealed that the helical polymer is an excellent chiral catalyst for cross-R–C reactions and is applicable to a wide range of substrates.

The amphiphilic poly(1s$_{50}$-$b$-2$_{100}$) copolymer had a higher $M_n$ than those of the reactants and products of the R–C reaction and exhibited excellent thermo-responsiveness in water. These characterizations facilitated not only product isolation but also polymer recovery and recycling. Thus, when the R–C reaction of **3a** and **4a** catalyzed by poly(1s$_{50}$-$b$-2$_{100}$) in water was accomplished, the aqueous solution was heated to 55 °C, higher than the cloud point of poly(1s$_{50}$-$b$-2$_{100}$). The transparent solution immediately turned turbid because of polymer precipitation (Fig. 3c). The precipitated solid was filtrated and washed completely using $n$-hexane to remove the residues of the product and unreacted substrates. The filtrate containing the R–C reaction product was purified and subjected to further analyses. The filter cake of the poly(1s$_{50}$-$b$-2$_{100}$) catalyst was reused in the cross-R–C reaction of **3a** and **4a**. To our delight, the recovered catalyst showed high catalytic activity and enantioselectivity. The yield and ee values of the $R$-**5aa** product using the recycled catalyst were almost the same as those generated using the fresh poly(1s$_{50}$-$b$-2$_{100}$) catalyst. Poly(1s$_{50}$-$b$-2$_{100}$) was recycled 10 times and maintained high activity and enantioselectivity (Fig. 3d). The yield and ee of the product $R$-**5aa** after the 10th reaction were 75% and 94%, respectively.

## Discussion

### Mechanism study

Because the reactants of the R–C reaction were insoluble in water, the enhanced activity and enantioselectivity of the catalytic block copolymer in water were ascribed to the hydrophobic core of the self-assembled micelles. The amphiphilic block copolymer was self-assembled into spherical micelles in water with hydrophobic and organocatalytic phosphine pendants at the interior and the hydrophilic poly-2$_n$ block at the exterior. The water-insoluble reactants were mainly located in the hydrophobic pocket of the micelles. The helical poly-1s$_m$ block bearing catalytic phosphine pendants at the interior provided not only catalytic phosphine for the R–C reaction but also a hydrophobic and asymmetric environment for enhancing the enantioselectivity. The asymmetric R–C reaction was catalyzed by the phenyl phosphine pendants. The one-handed helical backbone just provided an additional chiral environment and improved the enantioselectivity[31–36,56]. Thus, the reaction followed a mechanism similar to that of phenyl phosphine-catalyzed R–C reaction[50–54]. Moreover, the enriched local concentration of the water-insoluble reactants at the interior accelerated the reaction rate. Collectively, the synergistic effects of the self-assembled micelle facilitated the R–C reaction of water-insoluble materials in water and improved its activity and enantioselectivity. In order to obtain more information, the reaction of **3a** with **4a** catalyzed by poly(1s$_{50}$-$b$-2$_{100}$) in water was further monitored using $^{31}$P NMR spectroscopy. Clearly, there was no interaction between **3a** and poly(1s$_{50}$-$b$-2$_{100}$) as no change was observed on the $^{31}$P NMR spectrum (Supplementary Fig. 19). Meanwhile, an obvious interaction between **4a** and poly(1s$_{50}$-$b$-2$_{100}$) was observed because of a new appearance of a $^{31}$P peak at 33.26 ppm. The Raman analyses also

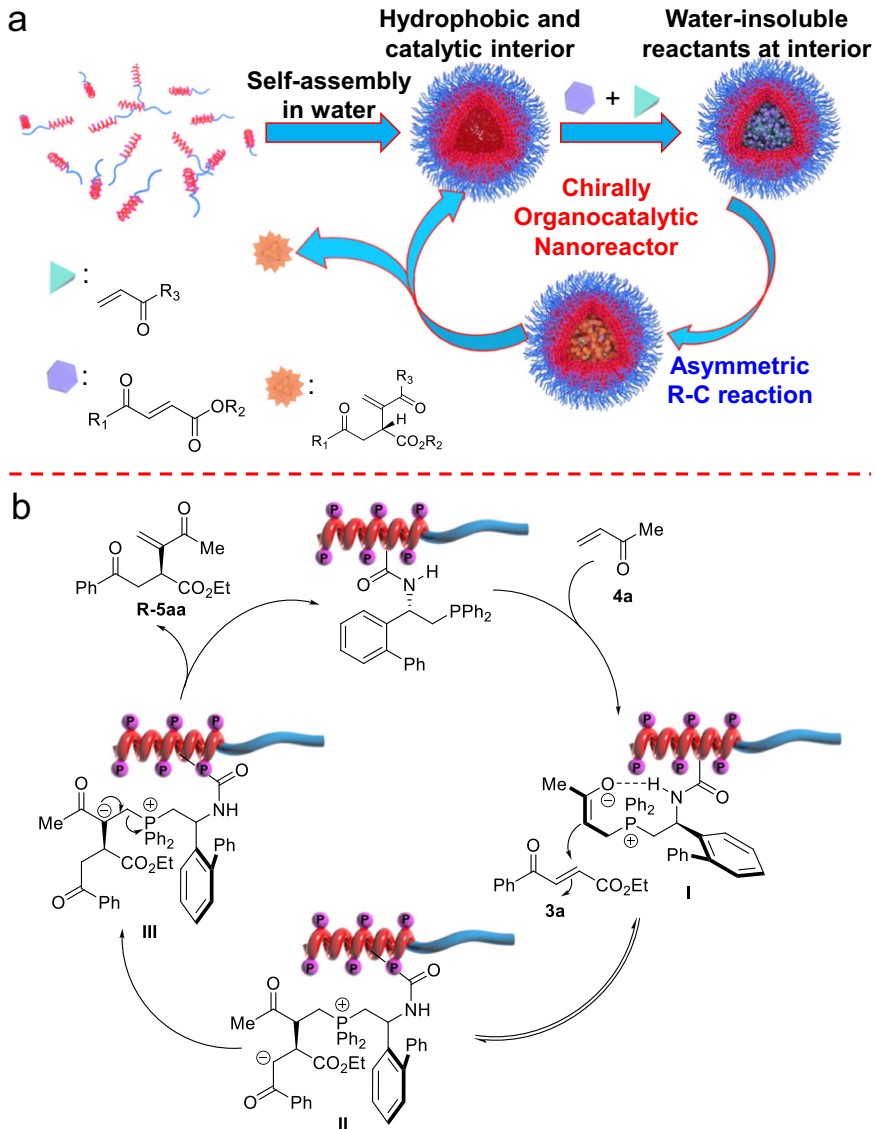

**Fig. 4 | Mechanism of the polymer-catalyzed R–C reaction. a** Schematic illustration of the helical polymer-micelle catalyzed asymmetric R–C reaction. **b** Possible mechanism for the polymer-catalyzed R–C reaction.

evidenced the intermolecular interaction between poly($1s_{50}$-$b$-$2_{100}$) with **4a** in water (Supplementary Fig. 20)[22]. Based on the above-mentioned results and previous works, we propose a possible mechanism for the reaction[52–54]. As shown in Fig. 4, the asymmetric cross R–C reaction was initiated by a Michael addition of the phosphine pendant to **4a** and gave an intermediate I. The subsequent nucleophilic addition of I to **3a** yielded intermediate II. Because of the asymmetric environment of the helical backbone, the addition mainly took place from the less steric side of the helix and thus gave intermediate II with high enantioselectivity. Following proton transfer, the final R–C product **R-5aa** and the phosphine catalyst were extruded from intermediate III. Another possible route was the elimination of β-H of intermediate II to obtain the target product. However, based on the work reported by Yu et al., the energy of migration followed by elimination is lower than that for the elimination of β-H in intermediate II[60]. Thus, migration followed by an elimination process is more likely to take place. The amide bonds on the pendants stabilized the one-handed helicity of the polyisocyanide backbone via intramolecular hydrogen bonding[17,18,55,56]. Moreover, the amide group contributed intramolecular hydrogen bonds and thus stabilized the intermediate I, which enhanced the enantioselectivity of the cross-R–C reaction[61].

To obtain more information about the R–C reaction, we conducted the poly($1s_{50}$-$b$-$2_{100}$) catalyzed reaction of deuterium-labeled **3a**-$d^1$ and **4e**-$d^2$ in $H_2O$, and the reaction of undeuterated **3a** with **4e** in deuterium water $D_2O$ according to above procedure (Fig. 5). The deuterium-labeled experiments gave the desired products with deuterium located at the expected positions. Meanwhile, the reaction conducted in $D_2O$ yielded the same product as that in $H_2O$. All these studies further confirmed the proposed mechanism.

In summary, we synthesized a family of amphiphilic helical poly-isocyanide block copolymers that self-assembled into well-defined chiral micelles in water with catalytic phosphine buried inside the hydrophobic pocket. Such an organocatalytic chiral micelle could efficiently catalyze asymmetric cross R–C reaction of various water-insoluble materials in water and deliver the desired products in high yields with excellent ee values. Here, the ee of the product reached 96% in >81% yields. Moreover, the enantioselectivity could be reversed using helical polyisocyanide copolymers possessing an opposite backbone helicity. The polymer catalysts were applicable to various reactants with just 4 mol% catalyst loading. Moreover, the block copolymers had excellent thermo-responsiveness in water with a cloud point of ~38.4 °C. Taking advantage of the thermo-responsiveness and

**Fig. 5 | Deuterium labeled experiments.** Asymmetric cross R–C reactions using **3a** and deuterium labeled **3a**-$d^1$ with deuterium-labeled **4e**-$d^2$ in $H_2O$, and **3a**-$d^1$ with **4e**-$d^2$ in deuterated water $D_2O$ at 0 °C with 4% of catalyst loading.

high $M_n$, the polymer catalyst was recycled 10 times with maintained its high reactivity and enantioselectivity. This study not only provides excellent and environment-friendly catalysts for asymmetric R–C reactions in water but also facilitates the exploration of green catalysts for producing chiral materials.

## Data availability
The synthetic details and experimental data generated in this study are all provided in the Supplementary Information. All other data are available from the corresponding author upon request.

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

## Acknowledgements

This work is supported by the Natural Science Foundation of China (NSFC, Nos. 92256201 (Z.-Q.W.), 52273204 (L.Z.), 52273006 (N.L.), 22071041 (Z.-Q.W.), 21971052 (N.L.), 51903072 (L.Z.), and 21871073 (Z.-Q.W.)) and the Fundamental Research Funds for the Central Universities. L. Zhou thanks Anhui Provincial Natural Science Foundation (Grant No. 2008085MB51).

## Author contributions

Z.-Q.W., N.L. and L.Z. designed and directed the project; L.X., Y.-X.L., and R.-T.G. performed the experiments and analyzed the data. Z.-Q.W., N.L. L.Z. and Z.C. wrote the manuscript with input from all other authors.

## Competing interests

The authors declare no competing interests.
