## [Peer Review File · Nature Communications]

Thermo-Responsive Chiral Micelles as Recyclable
Organocatalyst for Asymmetric Rauhut-Currier Reaction in
WaterReviewers' Comments:

Reviewer #1:

Remarks to the Author:

Wu and co-workers here present a thermo-responsive chiral micelles as recyclable organocatalyst for asymmetric Rauhut-Currier reaction in water. This manuscript (NCOMMS-22-47043) have synthesized a family of amphiphilic helical polyisocyanide block copolymers that self-assembled into well-defined chiral micelles in water with catalytic phosphine buried inside the hydrophobic pocket. This chiral micelle could be used as an efficiently catalyst for the asymmetric cross R-C reaction, it showed thermo-responsiveness with the lower critical solution temperature (LCST) of 38.4 °C. In the presence of 4 mol % catalyst loading, the RC reaction proceeded smoothly, and a variety of water-insoluble substrates are transferred to target products in high yield with excellent enantioselectivity. The yield and enantiomeric excess (ee) of the product generated in water is up to 90% and 96%, respectively. On the other hand, the polymer catalyst is easy to recover and has been recycled 10 times while maintaining activity and enantioselectivity. This work not only develops a new micelle catalyst, but also provides a asymmetric R-C reactions in water. For this reason, I believe that the current work merits publication in the Nat. Comm. after the following points have been addressed:

1. About the introduction, a brief introduction of the Rauhut-Currier (RC) reaction should be included.
2. About the reference,
 - 1). for reviews on the RC reaction: a) J. L. Methot, W. R. Roush, *Adv. Synth. Catal.* 2004, 346, 1035-1050; b) P. Xie, Y. Huang, *Eur. J. Org. Chem.* 2013, 6213-6226 should be cited.

Reviewer #2:

Comments:

The Wu and co-workers describe in this manuscript is firstly the synthesis of an amphiphilic helical polyisocyanide copolymers (using helical polyisocyanide bearing phosphine pendants as hydrophobic part and polyisocyanide carrying methyl triglycol chains as hydrophilic part), and then its self-assembly performance and application in cross R-C reaction. Both R-configured and S-configured polymer chains were prepared, and CD and UV-vis spectra were employed to confirm the chirality of polymer chain. Specifically, in an aqueous medium, the polymers could be self-assembled into well-defined chiral micelles and the hydrophobic phosphines are located in the interior. Catalytic reaction indicated that asymmetric cross R-C reaction of a variety of water-insoluble substrates could be well performed in water. The yields and ee's are up to 90% and 96%. In addition, the polymer could also be recovered and reused 10 times. From the overall innovation point view of the manuscript, it seems acceptable in Nat. Commun. However, too much important information is missing and mechanistic study can not support the conclusion of the manuscript. The criticisms are: 1) the concept of helicity of the polymer and chiral carbon center should be clearly distinguished. According to the visual presentation in Fig 1, both 1s and 1r contain a chiral carbon center. After first polymerization process, helical poly-1s and 1r were obtained. This is a very interesting polymerization process. However, I question the role of helicity of the polymer backbone. the author should present direct evidence that the enantioselectivity of R-C reaction was determined by chiral carbon center, or helicity of the polymer backbone, or the co-effect of the chiral carbon center and helicity of the polymer backbone. The author should provide the synergy mechanism if co-effect role exists; 2) two much less important information was presented in the introduction part, it will be better if the authors can rewrite the introduction part. For example, *“Although synthetic polymers, such as polystyrene, polydivinylbenzene, and polyacrylate derivatives, have been explored as supports for molecular catalysts, the catalytic activity and selectivity are generally decreased when compared with the corresponding molecular catalysts.”* This is not the truth, please see Chem. Rev. 2003, 103, 9, 3401–3430, Molecules. 2010; 15(9): 6306–6331. <https://www.sciencedirect.com/topics/chemistry/polymer-supported-reagent>. *“While the intermolecular cross R-C reaction still remains a challenging task.”* Several papers could be found by a simple search, please Rsc Adv. 2017, 7, 2890-2896; Adv. Synth. Catal. 2017, 359, 3934-3939; Org. Chem. Front. 2022, 9, 4840-4845. Overall, the present manuscript contains too many exaggerated sentences. For the SI part, I like the NMR spectra and HPLC analysis, the quality of these spectra is very high. However, a careful NMR analysis should be performed. If possible, those peaks from 3.0 ppm to 5.0 ppm should be enlarged as a butterfly hatch. Therefore, before I could further consider this manuscript for NC, the removal of exaggerated sentences and major revisions are required. Particularly I have serious concerns about the proposed mechanism in its current form that need to be addressed. Certain further control reactions are required.

Major revisions:

1. The catalytic data of corresponding homogeneous catalyst and 1r/1s were missing. Therefore, it is difficult to confirm the advantage of as-synthesized polymers. There is no sufficient evidence to confirm the superiority of as-synthesized polymers. Therefore, some control experiments must be added.
2. Generally, for oil substrates, it is possible to perform the catalytic reaction in pure water. For solid substrates, the addition of organic solvents (even little amount of co-solvent) is mandatory. However, in author's case, no organic solvent was added. How to solve the solubility problem of the solid substrate.
3. poly(1s50-b-2100) micelle was selected as the best catalyst to promote the cross R-C reaction in this manuscript. However, in SI part, the catalyst was prepared in 5.0 mg scale. This is totally not enough for catalytic reactions. Therefore, is it possible to synthesize the micelle in large scale? If yes, please also provide the characterization data of the final product.
4. For catalytic reactions, the authors should provide a detailed procedure. In addition, I question the amount of reaction solvent. In all case, 5 mL of reaction solvent was used. However, the local concentration of substrate around the micelle have an important influence on the catalytic result. There, a series of high-concentration reactions should be performed (from 0.2 mL of H₂O to 5 mL). the authors should carefully study the relationship between reaction concentration and yield/ee. May a concentration curve is needed.
5. The authors provide the characterization data of **5**. However, the NMR spectra of **3** and **4** is missing, even though they are known compounds. In addition, the HRMS, specific rotation and melting point are missing. These data must be added.
6. Have authors ever attempted a **3** functionalized by a heterocycle? If the R1 group of **3** was replaced by an aliphatic chain, what happen?
7. Utility7: How to determine the loading of the catalyst, by ICP or elementary analysis?
8. Due to the fact that the substrate of the R-C reaction water-insoluble, it seems that the catalytic reaction should occur within the hydrophobic core of the self-assembled micelles. However, for a high-level academic paper, direct evidence is mandatory. For the micelle, is there some mesoporous or microporous to allow for substrate exchange?
9. the author declared that: "*The helical poly-1sm block bearing catalytic phosphine pendants at the interior provides not only catalytic phosphine for the R-C reaction but also a hydrophobic and asymmetric environment for enhancing the enantioselectivity*". Generally, the enantioselectivity should be determined by the chiral center rather than the helical backbone. The authors should explain the relationship between enhanced enantioselectivity and the helical backbone.
10. a hydrogen bonding was presented in inter I, what will happen if the amide was protected by a methyl group? In addition, a series of deuterium labeling experiments should be added.
11. an anion transfer from int II to int III was proposed, I am interested in the energy

barriers of this process. Is it possible for Int II to undergo a phosphine β -hydride elimination process and followed by a protonation process in next step.

12. The description of ¹H NMR should be reanalyzed. For example, in the case of **5aa** (Note: there is the same problem in other cases), the peak around 4.18-4.08 is clearly not multiplet and, the J coupling constant is also not fully consistent (17.7 vs 18.0). In addition, the full spectra (from -1 ppm to 10 ppm should be given).
13. The HPLC spectra of R-5ea and 5fa have the reverse peak compared with other spectra, the author provides some convincing explanation for instance X-ray data.
14. Table 1, entry 4, the formula of solvent is incorrect.
15. For some point in the introduction part, for instance “Soluble polymers are less routinely used catalyst supports that could provide a solvent-like environment for organic reactions.”; “Helical polymers can be good skeletons to support chiral organocatalyst because the helical backbone provides additional chiral microenvironments, which can improve the stereoselectivity of an asymmetric reaction.” Please add Ref.
16. The author declares that: “Encouraged by these results, the helical polymer-based catalyst was applied to the cross-R-C reaction using aryl-substituted vinyl ketones (**4b-4d**), which have never been reported to date”. This is not the truth since the aryl-substituted vinyl ketones have been used in R-C reaction by Zhang group (please see <https://doi.org/10.1002/adsc.201700666>)
17. For the ¹³C NMR of those products containing F, the J coupling constant should be added.

Other mistakes (selected, not all):

‘While the intermolecular cross R-C reaction still remains a challenging task’

which suggest that the block copolymer has excellent thermo-responsiveness (Fig. 2a).

‘Gratifyingly, the reaction of **3a** and **4a** delivered the desired product *S*-**5aa** in 82% yield and -95% *ee*, which are similar to the results obtained using the poly(**1s**_{50-b}-**2**₁₀₀) catalyst with opposite enantioselectivity’

‘Collectively, these studies reveal that helical polymers are excellent chiral catalysts for cross R-C reactions and are applicable to a wide range of substrates.’

‘The transparent solution immediately turned turbid due to the precipitation of the polymer catalyst.

‘water insoluble’ to ‘water-insoluble’.

‘the final R-C product *R*-**5aa** and phosphine catalyst were extruded from intermediate III.’

Reviewer #3:

Remarks to the Author:

This article by the Wu group describes the synthesis of chiral micelles from poly(isocyanide) bearing phosphine pendants for asymmetric Rauhut-Currier reaction in water. In the presence of the chiral micelle, the product is afforded in high yields and enantiomeric excess (ee). The yields and ee values of the same Rauhut-Currier reaction catalyzed by the polymer itself in organic solvent, however, are significantly lower. The authors attribute the enhanced activity and enantioselectivity to the catalytic block copolymer to the hydrophobic micelle core. The polymers are characterized well via gel permeation chromatography and ¹H-NMR spectroscopy and the chirality of the poly(isocyanide) block is confirmed by circular dichroism spectroscopy in various solvents. Micelle formation has also been confirmed by atomic force microscopy (AFM) and transmission electron microscopy (TEM) and the critical micelle concentration value has been investigated. A substrate screen is conducted to demonstrate the scope of the chiral micelles. The mechanism by which the reaction proceeds is also elucidated. The block copolymer has a lower critical solution temperature at around 38 °C and this property is used to recycle and reuse the polymer for up to 10 cycles.

The main innovation of this manuscript is the following statement: 'Owing to the poor solubility of reactants and limited catalysts, efficiently cross R-C reaction in water with high enantioselectivity has not been realized to date. This study not only provides a series of excellent and environment-friendly catalysts for asymmetric R-C reactions in water but also paves a way for exploring novel green catalysts for producing chiral materials.' Chiral main-chain polymeric catalysts have previously been synthesized and reported (RSC Adv 2014, 4, 52023). Chiral micelles have also previously been prepared using polypeptides. This is, however, the first report of a poly(isocyanide) based chiral micelle. Additionally, polymeric chiral catalysts for R-C reaction in water is new with a wide range of substrates applied. The work reported here has the potential to be exciting and timely. However I cannot recommend the publication of this manuscript yet because (a) the idea of using chiral polymers as catalysts is not really new, (b) I do not think the 'micelles' are micelles and (c) a large number of control experiments are missing.

Detailed comments:

1. The following statement is not true: 'Although synthetic polymers, such as polystyrene, polydivinylbenzene, and polyacrylate derivatives, have been explored as supports for molecular catalysts, the catalytic activity and selectivity are generally decreased when compared with the corresponding molecular catalysts.' There are lots of examples where polymer supports increase rates.
2. The micelle size does not make sense. A micelle of 700 or 300 nm is way too large for the size of the amphiphilic polymers. A 20kD polymer assembled into micelles cannot yield a 700nm nanostructure. These are not micelles. I suggest to use cryo TEM to further characterize the nanostructures.
3. The authors need to confirm that the high yields and selectivity are due to the hydrophobic environment of the chiral micelle and not due to the presence of water that potentially causes chiral induction. A background conversion test needs to be conducted using poly(isocyanide) (without the hydrophilic block) in water.
4. Figure 1, please change the blue color of the P-helix. It is confusing to differentiate from the hydrophilic block. Consider changing the color of the poly(isocyanide) ChemDraw to black and label the 1s and 1r key with the same color as the polymer helix. Add the GPC eluent to the caption.
5. Please describe what instrument was used to get Fig. 2a in Supporting Information. In Fig. 2a, please indicate what solid line and dashed line represent respectively. How to determine the LCST of the polymers from Fig. 2a should also be mentioned in the paragraph.
6. In Fig. 2b, provide the concentration of micelle solution and polydispersity for each DLS trace.
7. Table 1, enantiomeric excess should not be reported as a negative value. Consider using R/S instead.
8. Can you provide the yield and enantioselectivity of unsupported phosphine catalyst for R-C reaction in water as a control experiment?

9. Among the family of amphiphilic polymers, only the morphology of poly(1s50-b-2100) was characterized in detail. Can you provide the DLS data of other polymers? This may help explain why poly(1s50-b-2100) showed the best results to catalyze R-C reaction.
10. Fig. 3a error bars are needed.
11. In the recycling part, what size of the filter was used? Please describe the recycling process details in Supporting Information.
12. How do you determine the repeating units ("n") of each polymer? The integration of peaks should be shown in NMR spectra in Supporting Information (Supplementary Fig.1, 3, 5, 7)
13. Round ^{13}C data to the nearest number after the period.
14. The mechanism part is pure speculation without a single scientific evidence. One can imagine a number of experiments to trap species, characterize species etc. This part either needs to be completely removed or substantiated partially with some data.

Dear reviewers,

We really appreciate the professional comments and suggestions. The manuscript was carefully revised to address each concern. The revisions we made in the manuscript were highlighted with red color. The point-by-point responses to the comments are as follows:

Response to Reviewer #1:

Comments: Wu and co-workers here present a thermo-responsive chiral micelles as recyclable organocatalyst for asymmetric Rauhut-Currier reaction in water. This manuscript (NCOMMS-22-47043) have synthesized a family of amphiphilic helical polyisocyanide block copolymers that self-assembled into well-defined chiral micelles in water with catalytic phosphine buried inside the hydrophobic pocket. This chiral micelle could be used as an efficiently catalyst for the asymmetric cross R-C reaction, it showed thermo-responsiveness with the lower critical solution temperature (LCST) of 38.4 °C. In the presence of 4 mol % catalyst loading, the RC reaction proceeded smoothly, and a variety of water-insoluble substrates are transferred to target products in high yield with excellent enantioselectivity. The yield and enantiomeric excess (ee) of the product generated in water is up to 90% and 96%, respectively. On the other hand, the polymer catalyst is easy to recover and has been recycled 10 times while maintaining activity and enantioselectivity. This work not only develops a new micelle catalyst, but also provides a asymmetric R-C reactions in water. For this reason, I believe that the current work merits publication in the Nat. Comm. after the following points have been addressed:

Response: We really appreciate the very positive comments.

1. About the introduction, a brief introduction of the Rauhut-Currier (RC) reaction should be included.

Response: Many thanks to the reviewer's suggestion. A brief introduction has been added in the introduction (page 3, lines 58-61): "The Rauhut-Currier (R-C) reaction of two active olefins is a unique and efficient approach for constructing carbon-carbon bonds and densely functionalized organic building blocks.⁵⁰⁻⁵⁴ Phosphine-catalyzed intermolecular cross R-C reaction is particularly intriguing among various organocatalyzed reactions.⁵²"

2. About the reference, 1). for reviews on the RC reaction: a) J. L. Methot, W. R. Roush, *Adv. Synth. Catal.* 2004, 346, 1035-1050; b) P. Xie, Y. Huang, *Eur. J. Org. Chem.* 2013, 6213-6226 should be cited.

Response: These important papers were cited in refs. 50 and 51 in the manuscript, we are sorry for the careless omissions.

Response to Reviewer #2:

Response to the comments:

1. The concept of helicity of the polymer and chiral carbon center should be clearly distinguished. According to the visual presentation in Fig 1, both **1s** and **1r** contain a chiral carbon center. After first polymerization process, helical poly-**1s** and **1r** were obtained. This is a very interesting polymerization process. However, I question the role of helicity of the polymer backbone. the author should present direct evidence that the enantioselectivity of R-C reaction was determined by chiral carbon center, or helicity of the polymer backbone, or the co-effect of the chiral carbon center and helicity of the polymer backbone. The author should provide the synergy mechanism if co-effect role exists.

Response: The polymer catalysts bear chiral carbon centers on the pendants and possess chiral helical backbone. Thus, to obtain details on the enantioselectivity, the asymmetric R-C reaction of **3a** with **4a** catalyzed by poly-**1s**₅₀ homopolymer and monomers **1r/1s** monomers were conducted under the identical conditions (runs 19-23, Table 1). The reactions in water gave **5aa** in 51% yield 69% *ee* using poly-**1s**₅₀ as catalyst, while it gave **5aa** in 57% yield with 48% *ee* by using **1s** as catalyst, respectively. The homogeneous reactions in CHCl₃ gave the desired **5aa** in 82% yield and 76% *ee* by poly-**1s**₅₀, and 78% yield and 57% *ee* by **1s**. The **1r** showed similar behaviors to **1s**, but in opposite enantioselectivity. The relatively higher yield and *ee* valued obtained by poly-**1s**₅₀ than those of **1s** confirm the synergistic effect of the helical backbone and the chiral pendants. We have added this result in the manuscript (page 8, lines 165-173).

2) two much less important information was presented in the introduction part, it will be better if the authors can rewrite the introduction part. For example, “*Although synthetic polymers, such as polystyrene, polydivinylbenzene, and polyacrylate derivatives, have been explored as supports for molecular catalysts, the catalytic activity and selectivity are generally decreased when compared with the corresponding molecular catalysts.*” This is not the truth, please see *Chem. Rev.* 2003, 103, 9, 3401–3430, *Molecules.* 2010; 15(9): 6306–6331. <https://www.sciencedirect.com/topics/chemistry/polymer-supported-reagent>. “*While the intermolecular cross R-C reaction still remains a challenging task.*” Several papers could be found by a simple search, please *Rsc Adv.* 2017, 7, 2890-2896; *Adv. Synth. Catal.* 2017, 359, 3934-3939; *Org. Chem. Front.* 2022, 9, 4840-4845. Overall, the present manuscript contains too many exaggerated sentences. For the SI part, I like the NMR spectra and HPLC analysis, the quality of these spectra is very high. However, a careful NMR analysis should be performed. If possible, those peaks from 3.0 ppm to 5.0 ppm should be enlarged as a butterfly hatch. Therefore, before I could further consider this manuscript for NC, the removal of exaggerated sentences and major revisions are required. Particularly I have serious concerns about the proposed

mechanism in its current form that need to be addressed. Certain further control reactions are required.

Response: Many thanks to the reviewer's suggestion. We have rewritten the introduction part, and the less important information was removed.

According to the suggestion, peaks from 3.0 ppm to 5.0 ppm regions on the ^1H NMR spectra in SI were enlarged as a butterfly hatch (Supplementary Fig. 20-32, Supplementary Fig. 34, Fig. 36, Fig. 38, Fig. 40, Fig. 42, Fig. 43, in SI).

For the mechanism study, a series of control experiments were performed, including the asymmetric R-C reaction using poly-**1s**₅₀ homopolymer, **1s/1r** monomer as catalysts. Moreover, deuteration experiments (Figure 5) and ^{31}P NMR spectra (Supplementary Fig. 18.) were also performed. All these studies further conform the proposed mechanism. Please refer to the responses to the following comments.

Response to major revisions:

1. The catalytic data of corresponding homogeneous catalyst and **1r/1s** were missing. Therefore, it is difficult to confirm the advantage of as-synthesized polymers. There is no sufficient evidence to confirm the superiority of as-synthesized polymers. Therefore, some control experiments must be added.

Response: To obtain details on the enantioselectivity, the asymmetric R-C reaction of **3a** with **4a** catalyzed by poly-**1s**₅₀ homopolymer and monomers **1r/1s** monomers were conducted under the identical conditions (runs 19-23, Table 1). The reactions in water gave **5aa** in 51% yield 69% *ee* using poly-**1s**₅₀ as catalyst, while it gave **5aa** in 57% yield with 48% *ee* by using **1s** as catalyst, respectively. The homogeneous reactions in CHCl_3 gave the desired **5aa** in 82% yield and 76% *ee* by poly-**1s**₅₀, and 78% yield and 57% *ee* by **1s**. The **1r** showed similar behaviors to **1s**, but in opposite enantioselectivity. The relatively higher yield and *ee* valued obtained by poly-**1s**₅₀ than those of **1s** confirm the synergistic effect of the helical backbone and the chiral pendants. We have added this result in the manuscript (page 8, lines 165-173).

2. Generally, for oil substrates, it is possible to perform the catalytic reaction in pure water. For solid substrates, the addition of organic solvents (even little amount of co-solvent) is mandatory. However, in author's case, no organic solvent was added. How to solve the solubility problem of the solid substrate.

Response: We are sorry for the careless negligence. Less than 5% volume of THF was used in case the substrates could not be dissolved in water. We have added an explanation in the manuscript (page 9, lines 180-181).

3. poly(**1s**₅₀-*b*-**2**₁₀₀) micelle was selected as the best catalyst to promote the cross R-C reaction in this manuscript. However, in SI part, the catalyst was prepared in 5.0 mg scale. This is totally not enough for catalytic reactions. Therefore, is it possible to

synthesize the micelle in large scale? If yes, please also provide the characterization data of the final product.

Response: The micelle can be easily prepared in large scale. We have revised the experimental procedure in SI, and the related characterization data were also provided in SI (page S5 in SI, lines 14-21; Supplementary Fig. 14-15, SI).

4. For catalytic reactions, the authors should provide a detailed procedure. In addition, I question the amount of reaction solvent. In all case, 5 mL of reaction solvent was used. However, the local concentration of substrate around the micelle has an important influence on the catalytic result. There, a series of high-concentration reactions should be performed (from 0.2 mL of H₂O to 5 mL). The authors should carefully study the relationship between reaction concentration and yield/ee. May a concentration curve is needed.

Response: Detailed procedures of the catalytic reactions were provided in SI (page S7, lines 2-14 in SI). Moreover, a series of high-concentration reactions were performed and the relationship between reaction concentration and yield/ee has been investigated in details (Supplementary Fig. 17, SI). We have also added a description in the manuscript to address this issue (page 9, lines 177-179): “Considering the local concentration of substrates within the micelle might influence the catalysis, experiments at different substrate concentrations were performed. The best results were obtained using 0.1 mmol of **3a** with 0.3 mmol of **4a** in water. A further increase in concentration caused precipitation, whereas the dilution of the substrates gave the product in lower yield.”

5. The authors provide the characterization data of **5**. However, the NMR spectra of **3** and **4** is missing, even though they are known compounds. In addition, the HRMS, specific rotation and melting point are missing. These data must be added.

Response: We are sorry for the careless omissions. The NMR spectra of **3** and **4** have been supplied in SI (Supplementary Fig. 46-Fig. 64). The HRMS, specific rotation, and melting point of the unknown compounds have been added in the revised SI (page S69-S71, Supplementary Figs. 104-108; page S11, line 1 from the bottom; page S12, line 7 from the bottom; page S13, line 12 from the bottom; page S14, line 2; page S15, line 1).

6. Have authors ever attempted a **3** functionalized by a heterocycle? If the R1 group of **3** was replaced by an aliphatic chain, what happen?

Response: Many thanks to the reviewer’s suggestion. The substrate **3** functionalized by heterocycle, 2-thienyl (**3l**), 2-Furyl (**3m**), and *tert*-butyl (**3n**) was used in R-C reaction. Good yield and high enantioselectivity of the target products **5la**, **5ma**, and **5na** were obtained. The results have been added to the revised manuscript and briefly discussed

(page 12, lines 220-224): “Furthermore, the substrate scope exploration suggested that poly(**1s**₅₀-*b*-**2**₁₀₀) was also an efficient catalyst for substrates containing heteroaryl groups and aliphatic chains. For example, the reaction of **3l**, **3m** and **3n** with **4a** gave the expected products **5la**, **5ma** and **5na** in good yield and high enantioselectivity (runs 16–18, Table 2). Collectively, these studies revealed that the helical polymer is an excellent chiral catalyst for cross-R-C reactions and is applicable to a wide range of substrates.”

7. Utility7: How to determine the loading of the catalyst, by ICP or elementary analysis?

Response: The loading of the catalyst was determined by elemental analysis. We have added a description in SI (page S7, lines 3-4, SI) and in the footnote of Table 1.

8. Due to the fact that the substrate of the R-C reaction water-insoluble, it seems that the catalytic reaction should occur within the hydrophobic core of the self-assembled micelles. However, for a high-level academic paper, direct evidence is mandatory. For the micelle, is there some mesoporous or microporous to allow for substrate exchange?

Response: The block copolymers were dynamically self-assembled into hollowed micelles, which facilitate the substrate exchange. We have added a description in the manuscript (page 7, lines 133-136): “The relatively large size of the micelles was ascribed to the formation of hollowed spherical micelles because of the distinct rigid and rod-like backbone of polyisocyanides.⁵⁸ The hollowed micelles might facilitate substrate exchange during the following asymmetric R-C reaction”.

9. the author declared that: “*The helical poly-1sm block bearing catalytic phosphine pendants at the interior provides not only catalytic phosphine for the R-C reaction but also a hydrophobic and asymmetric environment for enhancing the enantioselectivity*”. Generally, the enantioselectivity should be determined by the chiral center rather than the helical backbone. The authors should explain the relationship between enhanced enantioselectivity and the helical backbone.

Response: According to the reviewer’s suggestion, a series of control experiments were performed using poly-**1s**₅₀ homopolymer and **1s/1r** monomers as catalysts. The results indicated a synergistic effect of the helical backbone and the chiral pendants. We have added a description in the manuscript (page 8, lines 165-173; page 9, line 174-177): “The polymer catalysts carried chiral carbon centers on the pendants and possessed a chiral helical backbone. Thus, to obtain details on enantioselectivity, asymmetric R-C reaction of **3a** with **4a** catalyzed by the poly-**1s**₅₀ homopolymer and **1r/1s** monomers were conducted under identical conditions (runs 19-23, Table 1). The reactions in water gave **5aa** in 51% yield with 69% *ee* using poly-**1s**₅₀ as the catalyst, whereas it gave **5aa** in 57% yield with 48% *ee* using **1s** as the catalyst. The homogeneous reactions in CHCl₃ gave the desired **5aa** in 82% yield and 76% *ee* using poly-**1s**₅₀; and 78% yield and 57% *ee* using **1s**. The **1r** showed behaviors similar to those of **1s** but with opposite

enantioselectivity (run 21, Table 1). The relatively higher yield and *ee* values obtained by poly-**1s**₅₀ than those of **1s** confirmed the synergistic effect of the helical backbone and the chiral pendants. In addition, an increase in *M*_n of poly-**1s**_n could further improve the enantioselectivity of **5aa** until the DP of poly-**1s**_n reached 50 (Supplementary Fig. 16a). These results indicated the *M*_n-dependent helicity of the poly-**1s**_n backbone and further supported that the enhanced enantioselectivity came from the helical chirality of the polymer catalyst.⁵⁶

10. a hydrogen bonding was presented in inter I, what will happen if the amide was protected by a methyl group? In addition, a series of deuterium labeling experiments should be added.

Response: We really appreciate the reviewer's suggestion. We added an explanation in the manuscript (page 13, lines 250-251; and page 14, lines 252-253): "The asymmetric R-C reaction was catalyzed by the phenyl phosphine pendants. The one-handed helical backbone just provided an additional chiral environment and improved the enantioselectivity.^{31-36,56} Thus, the reaction followed a mechanism similar to that of phenyl phosphine catalyzed R-C reaction.⁵⁰⁻⁵⁴" Moreover, according to the reported work literature (*Chem. Commun.*, **2016**, 52, 7612–7615), the R-C reaction couldn't occur if the catalyst with a methyl group protected amide. The relevant description and literature have been added in the manuscript (page 14, lines 270-273): "The amide bonds on the pendants stabilized the one-handed helicity of the polyisocyanide backbone via intramolecular hydrogen bonding.^{17,18,55,56} Moreover, the amide group contributed intramolecular hydrogen bonds and thus stabilized the intermediate I, which enhanced the enantioselectivity of the cross-R-C reaction.⁶⁰".

The deuterium labeling experiments have been conducted and added in the revised manuscript (page 14, lines 273-276; and page 15, lines 277-278): "To obtain more information about the R-C reaction, we conducted the poly(**1s**_{50-b-2}₁₀₀) catalyzed reaction of deuterium labeled **3a-d**¹ and **4e-d**² in H₂O, and the reaction of undeuterated **3a** with **4e** in deuterium water D₂O according to above procedure (Figure 5). The deuterium labeled experiments gave the desired products with deuterium located at the expected positions. Meanwhile, the reaction conducted in D₂O yielded the same product as that in H₂O. All these studies further confirmed the proposed mechanism."

11. an anion transfer from int II to int III was proposed, I am interested in the energy 3 barriers of this process. Is it possible for Int II to undergo a phosphine β-hydride elimination process and followed by a protonation process in next step.

Response: Yes, it is possible that the Int II may undergo a phosphine β-hydride elimination process and followed by a protonation process. We have added a description in the manuscript (page 14, lines 267-270): "Another possible route was the elimination of β-H of intermediate II to obtain the target product. However, based on the work reported by Yu et al., the energy of migration followed by elimination is lower than that for the elimination of β-H in intermediate II (*J. Am. Chem. Soc.* **129**,

3470–3471 (2007)).⁶¹ Thus, migration followed by an elimination process is more likely to take place.”

12. The description of ¹H NMR should be reanalyzed. For example, in the case of **5aa** (Note: there is the same problem in other cases), the peak around 4.18-4.08 is clearly not multiplet and, the J coupling constant is also not fully consistent (17.7 vs 18.0). In addition, the full spectra (from -1 ppm to 10 ppm should be given).

Response: Many thanks to the reviewer’s suggestion. We have reanalyzed the ¹H NMR of the product **5aa** and others, the J coupling constant was corrected and the full spectra (from -1 ppm to 10 ppm) of all the compounds have been given (Supplementary Fig. 20- Fig. 32, Supplementary Fig. 34, Fig. 36, Fig. 38, Fig. 40, Fig. 42- Fig. 43, in SI).

13. The HPLC spectra of *R*-**5ea** and **5fa** have the reverse peak compared with other spectra, the author provides some convincing explanation for instance X-ray data.

Response: Many thanks to the reviewer’s suggestion. The configuration of the products *R*-**5ea** and *R*-**5fa** was confirmed by the works reported by Zhang’s group previously (*Angew. Chem. Int. Ed.* **2015**, *54*, 14853–14857). And the literature has been cited in the SI (page S72, ref. 5, SI).

14. Table 1, entry 4, the formula of solvent is incorrect.

Response: We are sorry for the careless mistake. We have corrected the mistake in Table 1, entry 4.

15. For some point in the introduction part, for instance “Soluble polymers are less routinely used catalyst supports that could provide a solvent-like environment for organic reactions.”; “Helical polymers can be good skeletons to support chiral organocatalyst because the helical backbone provides additional chiral microenvironments, which can improve the stereoselectivity of an asymmetric reaction.” Please add Ref.

Response: The references have been cited in manuscript as refs. 5-9; refs. 31 and 32).

16. The author declares that: “Encouraged by these results, the helical polymer-based catalyst was applied to the cross-R-C reaction using aryl-substituted vinyl ketones (**4b-4d**), which have never been reported to date”. This is not the truth since the aryl-substituted vinyl ketones have been used in R-C reaction by Zhang group (please see <https://doi.org/10.1002/adsc.201700666>).

Response: We have corrected the description in the manuscript (page 11, lines 216-217): “Encouraged by these results, the helical polymer-based catalyst was applied to the cross-R-C reaction using aryl-substituted vinyl ketones (**4b-4d**).”

17. For the ^{13}C NMR of those products containing F, the J coupling constant should be added.

Response: The J coupling constant for the ^{13}C NMR of the product containing F has been added in SI (page S13, lines 5-7 from the bottom; page S14, lines 7-10, in SI).

18. Other mistakes (selected, not all): ‘While the intermolecular cross R-C reaction still remains a challenging task’ which suggest that the block copolymer has excellent thermo-responsiveness (Fig. 2a).

‘Gratifyingly, the reaction of **3a** and **4a** delivered the desired product **S-5aa** in 82% yield and -95% *ee*, which are similar to the results obtained using the poly(**1s50-b-2100**) catalyst with opposite enantioselectivity’

‘Collectively, these studies reveal that helical polymers are excellent chiral catalysts for cross R-C reactions and are applicable to a wide range of substrates.’

‘The transparent solution immediately turned turbid due to the precipitation of the polymer catalyst.

‘water insoluble’ to ‘water-insoluble’.

‘the final R-C product **R-5aa** and phosphine catalyst were extruded from intermediate III.’

Response: Many thanks to the reviewer’s suggestion. We have corrected all the mistakes and the English of manuscript was further polished by an English-native expert.

Response to Reviewer #3:

1. The following statement is not true: ‘Although synthetic polymers, such as polystyrene, polydivinylbenzene, and polyacrylate derivatives, have been explored as supports for molecular catalysts, the catalytic activity and selectivity are generally decreased when compared with the corresponding molecular catalysts.’ There are lots of examples where polymer supports increase rates.

Response: Many thanks to the reviewer’s suggestion. We have removed the inappropriate statement in the revised manuscript.

2. The micelle size does not make sense. A micelle of 700 or 300 nm is way too large for the size of the amphiphilic polymers. A 20kD polymer assembled into micelles cannot yield a 700 nm nanostructure. These are not micelles. I suggest to use cryo TEM to further characterize the nanostructures.

Response: We have added an explanation in the manuscript (page 7, lines 133-139). ‘The relatively large size of the micelles was ascribed to the formation of hollowed spherical micelles because of the distinct rigid and rod-like backbone of

polyisocyanides (*Macromolecules* **49**, 110–119 (2016)).⁵⁸ The hollowed micelles might facilitate substrate exchange during the following asymmetric R-C reaction. The cryo TEM image of poly(**1s**_{50-b-2}**100**) in water also supported the formation of spherical micelles with a diameter that agrees with that in the DLS analysis (Supplementary Fig. 15).”

3. The authors need to confirm that the high yields and selectivity are due to the hydrophobic environment of the chiral micelle and not due to the presence of water that potentially causes chiral induction. A background conversion test needs to be conducted using poly(isocyanide) (without the hydrophilic block) in water.

Response: According to the reviewers' comments, we did controlled experiments using poly-**1s**_m as catalysts reaction in water. The experimental results showed a lower yield and selectivity than the results of the chiral micelle catalyst. The relevant descriptions have been added in the manuscript (page 8, lines 165-173; page 9, lines 174-177): “The polymer catalysts carried chiral carbon centers on the pendants and possessed a chiral helical backbone. Thus, to obtain details on enantioselectivity, asymmetric R-C reaction of **3a** with **4a** catalyzed by the poly-**1s**₅₀ homopolymer and **1r/1s** monomers were conducted under identical conditions (runs 19-23, Table 1). The reactions in water gave **5aa** in 51% yield with 69% *ee* using poly-**1s**₅₀ as the catalyst, whereas it gave **5aa** in 57% yield with 48% *ee* using **1s** as the catalyst. The homogeneous reactions in CHCl₃ gave the desired **5aa** in 82% yield and 76% *ee* using poly-**1s**₅₀; and 78% yield and 57% *ee* using **1s**. The **1r** showed behaviors similar to those of **1s** but with opposite enantioselectivity (run 21, Table 1). The relatively higher yield and *ee* values obtained by poly-**1s**₅₀ than those of **1s** confirmed the synergistic effect of the helical backbone and the chiral pendants. In addition, an increase in *M*_n of poly-**1s**_n could further improve the enantioselectivity of **5aa** until the DP of poly-**1s**_n reached 50 (Supplementary Fig. 16a). These results indicated the *M*_n-dependent helicity of the poly-**1s**_n backbone and further supported that the enhanced enantioselectivity came from the helical chirality of the polymer catalyst.⁵⁶”

4. Figure 1, please change the blue color of the *P*-helix. It is confusing to differentiate from the hydrophilic block. Consider changing the color of the poly(isocyanide) ChemDraw to black and label the **1s** and **1r** key with the same color as the polymer helix. Add the GPC eluent to the caption.

Response: We have revised the colors of the structures and the GPC eluent was added in the caption of Figure 1 (page 5, Figure 1).

5. Please describe what instrument was used to get Fig. 2a in Supporting Information. In Fig. 2a, please indicate what solid line and dashed line represent respectively. How to determine the LCST of the polymers from Fig. 2a should also be mentioned in the paragraph.

Response: Thermo-responsive experiments (Fig. 2a) were carried on UNIC 4802 UV/vis double beam spectrophotometer instrument using transmission model. The experimental details have been added in SI (page S3, lines 1-2 from the bottom).

The solid lines and dashed lines in Fig. 2a in the main text indicate heating and cooling processes. A description has been added in the caption of Fig.2.

The LCST of the polymers from Fig. 2a determined from the temperature corresponding to 90% transmittance of the anti-sigmoidal transmittance–temperature curve during the heating process. The relevant description has been added in the revised manuscript (page 6, lines 117-118).

6. In Fig. 2b, provide the concentration of micelle solution and polydispersity for each DLS trace.

Response: The concentration (0.2 mg/mL) of micelle solution and polydispersity for each DLS trace were added in the caption of Fig. 2b. The concentrations were 0.2 mg/mL. The polydispersity values for the DLS analyses are 0.387 (H₂O), 0.235 (H₂O/THF = 75/25), 0.241 (H₂O/THF = 50/50), 0.312 (H₂O/THF = 25/75), and 0.222 (THF), respectively (page 7, lines 144-146).

7. Table 1, enantiomeric excess should not be reported as a negative value. Consider using R/S instead.

Response: According to the reviewers' suggestions, we have changed the negative value to *S* configuration in Table 1 (runs 18 and 21, in Table 1).

8. Can you provide the yield and enantioselectivity of unsupported phosphine catalyst for R-C reaction in water as a control experiment?

Response: According to the reviewers' suggestions, the controlled experiments using unsupported phosphine catalysts **1s** and **1r** were performed in water. The results have been added in Table 1 (runs 20, 21, and 23 in Table 1). We also added a description in the manuscript (page 8, lines 166-172): “Thus, to obtain details on enantioselectivity, asymmetric R-C reaction of **3a** with **4a** catalyzed by the poly-**1s**₅₀ homopolymer and **1r/1s** monomers were conducted under identical conditions (runs 19-23, Table 1). The reactions in water gave **5aa** in 51% yield with 69% *ee* using poly-**1s**₅₀ as the catalyst, whereas it gave **5aa** in 57% yield with 48% *ee* using **1s** as the catalyst. The homogeneous reactions in CHCl₃ gave the desired **5aa** in 82% yield and 76% *ee* using poly-**1s**₅₀; and 78% yield and 57% *ee* using **1s**. The **1r** showed behaviors similar to those of **1s** but with opposite enantioselectivity (run 21, Table 1).”

9. Among the family of amphiphilic polymers, only the morphology of poly(**1s**₅₀-*b*-**2**₁₀₀) was characterized in detail. Can you provide the DLS data of other polymers? This may help explain why poly(**1s**₅₀-*b*-**2**₁₀₀) showed the best results to catalyze R-C reaction.

Response: Many thanks to the reviewer's suggestion, the DLS data of other polymers has been measured and the data were provided in SI (page S24, Supplementary Fig.16, SI). We also added a description in the manuscript (page 7, lines 138-139): "Other block copolymers showed similar self-assembly properties in water, as revealed by the DLS analyses (Supplementary Fig. 16)."

10. Fig. 3a error bars are needed.

Response: According to the suggestion, the error bars have been added in Fig 3a (page 11, Figure 3a).

11. In the recycling part, what size of the filter was used? Please describe the recycling process details in Supporting Information.

Response: The precipitated polymer catalyst was filtrated by medium flow qualitative filter paper with bore diameter of 30-50 μm . The detailed recycling process was added in SI (page S7, lines 10-11).

12. How do you determine the repeating units ("n") of each polymer? The integration of peaks should be shown in NMR spectra in Supporting Information (Supplementary Fig.1, 3, 5, 7)

Response: Because the polymerizations proceeded in a well-controlled living polymerization mechanism, the degree of polymerization was consistent with the initial monomer-to-catalyst feed ratio, according to the detailed studies we reported previously.^{11,17,55,56} We have added an explanation in the manuscript (page 4, lines 89-91). The integration of peaks in NMR spectra (Supplementary Fig.1, 3, 5, 7) in SI were provided.

13. Round ^{13}C data to the nearest number after the period.

Response: According to the reviewer's suggestion, we have revised the ^{13}C data in SI. See page S12, lines 5-7 and lines 1-2 from the bottom; page S13, lines 1-2 from the top and lines 5-7 from the bottom; page S14, lines 7-10 from the top; page S15, lines 6-8 from the top in SI.

14. The mechanism part is pure speculation without a single scientific evidence. One can imagine a number of experiments to trap species, characterize species etc. This part either needs to be completely removed or substantiated partially with some data.

Response: Many thanks to the reviewer's suggestion. The asymmetric R-C reaction was catalyzed by the phenyl phosphine pendants. The one-handed helical backbone just provides additional chiral environment and improves the enantioselectivity.^{31-36,56} Thus the reaction followed a similar mechanism of phenyl phosphine catalyzed R-C

reaction.⁵⁰⁻⁵⁴ In order to obtain more information, the reaction of **3a** with **4a** catalyzed by poly(**1s**_{50-b}-**2**₁₀₀) in water was monitored by ³¹P NMR spectroscopy (Supplementary Fig. 18). Moreover, the control experiments using deuterium labeled **3a-d**¹ and **4e-d**² in H₂O, and undeuterated **3a** with **4e** in deuterium water D₂O were performed (Figure 4c). The deuterium labeled experiments gave the desired products with deuterium located at the expected positions. While the reaction carried out D₂O yielded the same product to that in H₂O. All these studies further confirmed the proposed mechanism. We have added explanations in the manuscript (page 13, lines 250-251; page 14, lines 252-253, lines 256-260, lines 267-276; page 15, lines 277-278).

Thanks again for all the professional and important opinions. We hope that this revised manuscript with improved quality can meet the high standards of *Nature Communications*.

Best regards,

Zong-Quan Wu

E-mail: zqwu@jlu.edu.cn (Z. W.)

State Key Laboratory of Supramolecular Structure and Materials, College of Chemistry, Jilin University, Changchun 130012, China.

Reviewers' Comments:

Reviewer #2:

Remarks to the Author:

The author had addressed the all questions, and it is suitable for publication

Reviewer #4:

Remarks to the Author:

Authors have responded to the concerns of the reviewer 3 with necessary reversion in the manuscript. However, some responses are not adequate and further revision is suggested. Specifically, (1) regarding the large size of the micelles formed, it's true that rigid polymers often produce aggregates with large sizes. However, micelles with size of 700 nm from DLS measurements is not really suitable, since DLS measurements cannot guarantee the accuracy for such large sizes due to multiple scattering problem. Furthermore, revised data from cryo-TEM are not suitable to compare to these from DLS measurements, since two methods are based on completely two different concepts. Their agreement itself can be a problem. Authors should be aware and check this carefully. Furthermore, checking mesh size in the large micelles may be helpful in understanding their formation and the following results in catalysis. (2) Regarding description and experiments on the thermoresponsive behavior of the polymers, the data are only support to address the turbid or cloud point, not LCST. Furthermore, thermoresponsive behavior is a dehydration and collapse process for a polymer during heating or cooling. So, it's not suitable to say "thermoresponsive experiments", instead, just "turbidimetry measurements". The cloud point for a polymer is normally defined as the point when transmittance of the aqueous solution reduced to 50% of its original value, but not 90%. (3) Regarding the mechanism, one-handed helical backbone provides additional chiral environment and improves the enantioselectivity, which is true only when the interactions between the chiral polymer substrates and reactants (here, phenyl phosphine) are strong enough. Therefore, authors should rephrase this description, better with Raman data to support their point view in answering the reviewer 3's concern.

Dear reviewer,

We really appreciate the professional comments and suggestions. The manuscript was carefully revised to address each concern. The revisions we made in the manuscript were highlighted with red color. The point-by-point responses to the comments and suggestions are as follows:

Response to Reviewer #4:

Comments: Authors have responded to the concerns of the reviewer 3 with necessary revision in the manuscript. However, some responses are not adequate and further revision is suggested. Specifically, (1) regarding the large size of the micelles formed, it's true that rigid polymers often produce aggregates with large sizes. However, micelles with size of 700 nm from DLS measurements is not really suitable, since DLS measurements cannot guarantee the accuracy for such large sizes due to multiple scattering problem. Furthermore, revised data from cryo-TEM are not suitable to compare to these from DLS measurements, since two methods are based on completely two different concepts. Their agreement itself can be a problem. Authors should be aware and check this carefully. Furthermore, checking mesh size in the large micelles may be helpful in understanding their formation and the following results in catalysis.

Response: We really appreciate the reviewer's suggestions. We have carefully re-measured the DLS of poly(**1s₅₀-b-2₁₀₀**). Based on the retested DLS data, the size of poly(**1s₅₀-b-2₁₀₀**) in THF/H₂O (v/v = 5/5) was 506 nm with the polydispersity of 0.341. We have revised the manuscript (page 6, lines 13-14): "After adding water, the diameters were 506 and 280 nm for the water contents of 50% and 75%, respectively." Furthermore, we have revised the description on cryo-TEM and deleted the comparison of cryo-TEM with DLS analyses (page 6, line 1 from the bottom; and page 7, lines 1-2): "The cryo-TEM image of poly(**1s₅₀-b-2₁₀₀**) in water also supported the formation of spherical micelles with a diameter of ca. 90 nm (Supplementary Fig. 14)."

(2) Regarding description and experiments on the thermoresponsive behavior of the polymers, the data are only support to address the turbid or cloud point, not LCST. Furthermore, thermoresponsive behavior is a dehydration and collapse process for a polymer during heating or cooling. So, it's not suitable to say "thermoresponsive experiments", instead, just "turbidimetry measurements". The cloud point for a polymer is normally defined as the point when transmittance of the aqueous solution reduced to 50% of its original value, but not 90%.

Response: According to the reviewer's suggestion, we have revised the manuscript (page 6, lines 4-10): "Interestingly, the transparent aqueous solution of poly(**1s**₅₀-*b*-**2**₁₀₀) turned turbid upon heating and became transparent again after cooling to room temperature, suggesting the turbidimetry-responsiveness with temperature (Fig. 2a). Detailed UV-vis absorption studies revealed that the cloud point was 38.4 °C for poly(**1s**₅₀-*b*-**2**₁₀₀), determined from the temperature corresponding to 50% transmittance of the antisigmoidal transmittance-temperature curve during the heating process.⁵⁷ The cloud point decreased with the elongation of poly-**2**_n block of the copolymers, it was 46.7 °C, 38.4 °C, and 30.0 °C for poly(**1s**₅₀-*b*-**2**₅₀), poly(**1s**₅₀-*b*-**2**₁₀₀), and poly(**1s**₅₀-*b*-**2**₁₅₀), respectively (Fig. 2a)."

(3) Regarding the mechanism, one-handed helical backbone provides additional chiral environment and improves the enantioselectivity, which is true only when the interactions between the chiral polymer substrates and reactants (here, phenyl phosphine) are strong enough. Therefore, authors should rephrase this description, better with Raman data to support their point view in answering the reviewer 3's concern.

Response: We really appreciate the reviewer's suggestion. The Raman analysis was conducted on poly(**1s**₅₀-*b*-**2**₁₀₀) with the presence of **3a** and **4a**. It clearly revealed that there was an intermolecular interaction between poly(**1s**₅₀-*b*-**2**₁₀₀) with **4a**. We have added a description in the manuscript (page 14, lines 9-10): "The Raman analyses also

evidenced the intermolecular interaction of between poly(**1s**_{50-b-2100}) with **4a** in water (Supplementary Fig. 20).”

Thank you again for all the professional and important opinions. We hope that this revised manuscript with improved quality can meet the high standards of *Nature Communications*.

Best regards,

Zong-Quan Wu

Reviewers' Comments:

Reviewer #4:

Remarks to the Author:

Authors have addressed my concerns properly, and the manuscript in its present state is suitable for publishing.